# Foundational Movement Skills and Play Behaviors during Recess among Preschool Children: A Compositional Analysis

**DOI:** 10.3390/children8070543

**Published:** 2021-06-24

**Authors:** Lawrence Foweather, Matteo Crotti, Jonathan D. Foulkes, Mareesa V. O’Dwyer, Till Utesch, Zoe R. Knowles, Stuart J. Fairclough, Nicola D. Ridgers, Gareth Stratton

**Affiliations:** 1Research Institute for Sport & Exercise Sciences, Liverpool John Moores University, Liverpool L3 3AT, UK; M.Crotti@2016.ljmu.ac.uk (M.C.); J.D.Foulkes@ljmu.ac.uk (J.D.F.); modwyer@earlychildhoodireland.ie (M.V.O.); Z.R.Knowles@ljmu.ac.uk (Z.R.K.); 2Institute of Educational Sciences, University of Münster, 48149 Münster, Germany; till.utesch@uni-muenster.de; 3Department of Sport and Physical Activity, Edge Hill University, Ormskirk L39 4QP, UK; faircls@edgehill.ac.uk; 4Institute for Physical Activity and Nutrition, School of Exercise and Nutrition Sciences, Deakin University, Geelong 3220, Australia; nicky.ridgers@deakin.edu.au; 5College of Engineering, Swansea University, Swansea SA2 8PP, UK; G.Stratton@swansea.ac.uk

**Keywords:** motor skills, fundamental movement skills, play, preschool, early childhood education centers, physical literacy, young children, early childhood, cross-sectional, observational

## Abstract

This study aimed to examine the associations between play behaviors during preschool recess and foundational movement skills (FMS) in typically developing preschool children. One hundred and thirty-three children (55% male; mean age 4.7 ± 0.5 years) from twelve preschools were video-assessed for six locomotor and six object-control FMS using the Champs Motor Skill Protocol. A modified System for Observing Children’s Activity and Relationships during Play assessed play behaviors during preschool recess. Associations between the composition of recess play behaviors with FMS were analyzed using compositional data analysis and linear regression. Results: Relative to time spent in other types of play behaviors, time spent in play without equipment was positively associated with total and locomotor skills, while time spent in locomotion activities was negatively associated with total and locomotor skills. No associations were found between activity level and group size play behavior compositions and FMS. The findings suggest that activity type play behaviors during recess are associated with FMS. While active games without equipment appear beneficial, preschool children may need a richer playground environment, including varied fixed and portable equipment, to augment the play-based development of FMS.

## 1. Introduction

Early childhood is recognized as a critical period for the development of foundational movement skills (FMS) [1,2,3]. FMS is a relatively new term that includes both traditionally conceptualized *fundamental* movement skills such as stability (e.g., sitting, standing, balancing on a foot), locomotor (e.g., running, jumping, crawling), and object control (e.g., striking, catching, throwing) skills, as well as other skills that support lifelong engagement in physical activity (e.g., squatting, cycling, swimming) [1]. ‘Foundational’ refers to these skills providing an ‘underlying base or support’, with the development of greater competency in many skills providing more options for physical activity across the life course [1]. FMS are typically poorly developed in preschool children as they find themselves at the rudimentary stage of development [4,5]. Young children will not acquire proficiency in FMS through growth and maturation alone: the rate and extent of FMS development is dependent on the interplay between environmental (e.g., access to equipment) and individual (e.g., confidence) factors [6,7,8] and is therefore non-linear and idiosyncratic [9]. Developing proficiency in these skills through childhood is important, as FMS provide an underlying base or support for successful participation in physical activities and sport across the life course [1,7,10,11,12]. For example, preschool children with higher FMS competence are more likely to have higher physical fitness and physical activity levels later in life [13,14]. Furthermore, accumulative evidence highlights the beneficial effects of FMS competence on wider aspects of child development. FMS have been linked to key elements of school readiness [15], including cognitive [16,17], language [18], and social [19] outcomes. Moreover, FMS level has been found to be inversely related with body mass index (BMI) [20,21] and positively associated with physical activity behaviors in preschool children [22,23,24].

Play is suggested to be an important context for FMS in the early years [9,25,26,27,28] and is characterized by activities that are freely chosen, self-directed, intrinsically motivated, and free from many constraints of objective reality [29,30,31]. The proportion of time spent in unstructured play reaches its peak in the preschool period before declining rapidly in the primary school years [31]. Through play, young children have opportunities to explore their environment and develop and practice FMS [9,25,26,27]. Play has also been noted for its potential to foster children’s strength, endurance, cognition, and prosocial behaviors, which in turn could mediate FMS development [28,31,32,33,34,35]. Active play [36], risky play [37], and outdoor play [38,39] are considered particularly beneficial for young children’s physical development, challenging their movement abilities such as balance, agility, coordination, and spatial awareness, as well as nurturing physical activity behaviors through activities that conjure up feelings of thrill and excitement. Despite these assertions, empirical studies examining the associations between FMS competence and play behaviors are lacking [40,41]. Indeed, studies to date have focused on environmental factors such as playground size rather than what play behaviors children are engaged in within play settings and with whom [40,41]. Such evidence could be used to identify what play behaviors could be targeted in FMS interventions for preschoolers.

An important context where many young children spend a significant proportion of their time is at preschool (i.e., early education and childcare settings such as kindergartens, nurseries, day care centers, and preschools—both public and private). In Western Europe and including the United Kingdom, over 90% of three-to-five-year-old children are enrolled at preschool [42]. In England, all three- and four-year-old children receive 15 h of free preschool education for 38 weeks of the year. English educational settings follow the Early Years Foundation Stage curriculum [43], which has emphasised play-based learning and development in several core areas including physical development, personal, social, and emotional development, and communication and language, among others. At preschool, young children can foster physical development through unstructured and outdoor free play during several regularly scheduled break times each day (recess periods). Increased levels of moderate-to-vigorous intensity physical activity within recess periods indicates that this may be an important environment for play and FMS development in a preschooler’s day [44]. A small positive relationship between children’s overall FMS competence and the playground size in the preschool setting has also been noted [45]. Whilst this latter study examined several preschool environmental characteristics, to the best of our knowledge, no study has examined the relationship between FMS and young children’s play behaviors during preschool recess.

The aim of the present study, therefore, was to examine the associations between play behaviors and FMS in typically developing preschool children during recess at preschool. Play behaviors during recess occur in a finite time window and are therefore mutually exclusive and co-dependent on each other (i.e., time spent in one behavior can only be changed by increasing or reducing time spent in at least one of the other play behaviors by the same duration). Thus, play behaviors should be analyzed and interpreted relative to one another as opposed to in isolation [46,47]. The present study therefore used compositional data analysis (CoDA) as a statistical approach for the inferential analysis of recess play behavior data against FMS outcomes [46,47]. CoDA is increasingly used in the field and robust to issues such as collinearity [47]. To our knowledge, this is the first study to use CoDA to analyze the association between recess play behaviors and FMS.

## 2. Methods

### 2.1. Design

This research was part of the Active Play project, which is described in detail elsewhere [44] and was approved by the University Ethics Committee (ref. 09/SPS/027). In summary, Active Play consisted of a 6-week educational programme conducted during class time that involved staff and children from preschools within disadvantaged communities and targeted children’s physical activity levels, FMS, fitness, and self-confidence. The data used in this cross-sectional study were collected through two phases of baseline assessments conducted during October 2009 and March 2010 to maximize recruitment and control for the influence of seasonal effects.

### 2.2. Settings and Participants

Twelve preschools situated in a large urban city in Northwest England and located within neighborhoods within the highest 10% for national deprivation [48] were randomly selected and invited to participate in the study. All of the preschools provided informed gatekeeper consent to participate. All children aged three- to- five-years-old at the study preschools were invited to participate and were required to return informed written parental consent, demographic information (home postcode, the child’s ethnicity and date of birth, and the mother’s highest level of education) and medical assessment forms. From the 673 eligible children, parental consent was obtained for 240 children (35% response rate). No children had any known medical conditions that could affect motor proficiency or participation in physical activity.

### 2.3. Measures

#### 2.3.1. Foundational Movement Skills

The Children’s Activity and Movement Assessment Study (CHAMPS) Motor Skill Protocol (CMSP) was used to assess the preschoolers’ FMS [49]. CMSP is a valid and reliable tool developed for three-to-five-year-old children and assesses process characteristics in six locomotor (run, broad jump, leap, hop, gallop, and slide) and six object-control (overarm throw, stationary strike, kick, catch, underhand roll, and stationary dribble) skills [49]. Following a single demonstration of each skill by a trained research assistant, children performed two trials of each skill in a standardized order while working in small groups of 2–4 children. Trials took place at preschool within either indoor halls or on outdoor playgrounds, depending on available facilities, and were recorded using a tripod-mounted video camera for later analysis. Skill components were subsequently marked as present (scored 1) or absent (scored 0) against the process criteria (e.g., arms extend forwards and upwards in the horizontal jump) by a single trained assessor [49]. Inter-rater reliability with an experienced assessor was established prior to assessment using pre-coded videotapes of 10 children, with 83.9% agreement across the twelve FMS (range 72.9–89.3%). The total number of skill components checked as present over two trials was summed to give a composite total skill score (possible range: 0–142 skill components), whilst locomotor (0–64 skill components) and object-control (0–78 skill components) subtest scores were created by summing the scores of skills within each subscale.

#### 2.3.2. Play Behaviors

A modified version of the System for Observing Children’s Activity and Relationships during Play (SOCARP) was used to assess preschool children’s play behaviors [50]. SOCARP is a validated tool designed to simultaneously assess multiple aspects of play using time sampling techniques where a 10 s observation period is followed by a 10 s recording period [50]. SOCARP codes play behaviors in four categories: *activity level* (lying, sitting, standing, walking, and very active), *group size* (alone, small group of 2–4 individuals, medium group of 5–9 individuals, and large group of ≥10 individuals), *activity type* (sport (e.g., football, tennis]), active games (e.g., dancing, throwing, and catching), sedentary (e.g., reading, artwork) and locomotion (e.g., running/walking/jogging/skipping that is not part of an active game, such as transitioning from one activity to another), and *interactions* (no interaction, physical sportsmanship, verbal sportsmanship, physical conflict, verbal conflict, and ignore). For this study, SOCARP was modified to enable a more detailed examination of play behaviors of young children. Specifically, the ‘sport’ category was removed from the *activity type* variable and the ‘active games’ category was divided into two categories: ‘active games with equipment’ (fixed (e.g., climbing frame) or portable/loose parts (e.g., balls or socio-dramatic props such as teacups)) or ‘active games without equipment’ (e.g., chasing games/rough and tumble). Furthermore, the ‘sedentary’ *activity type* category was divided into ‘sedentary’ (i.e., non-play sedentary behaviors, e.g., viewing others’ games but sitting as a spectator) and ‘quiet play’ (i.e., play-based sedentary behaviors, e.g., sitting playing board games). Each child was filmed for 5 min by a research assistant during a single morning (~20 min), lunch (~45 min), or afternoon (~20 min) recess period, which took place outdoors on the preschool play area. For each 10 s recording period, play behaviors across each category were coded, and the number of intervals (units) per behavior category were summed for use in the statistical analysis. A trained observer (MOD) retrospectively coded the play behaviors using video recordings. Observer training was conducted through coding a prerecorded sample of recess play videos, and >80% inter-rater agreement was obtained with an expert assessor (NR) for all SOCARP categories. Social interaction data was not included in the analyses due to the lack of a plausible conceptual association with FMS.

#### 2.3.3. Anthropometrics and Demographics

Body mass (to the nearest 0.1 kg) and stature (to the nearest 0.1 cm) were measured by trained researchers using digital scales and a portable stadiometer, respectively. BMI (kg/m^2^) was calculated and converted to zBMI using the ‘LMS’ method for analysis [51]. Information about children’s demographics (i.e., date of birth, gender, ethnicity, home postcode) were provided by parents or guardians within a questionnaire that was returned with the signed consent form. Household postcode was used to classify children into deciles of deprivation level using the English indices of deprivation [48].

### 2.4. Statistical Analysis

All statistical analysis was undertaken using R open-source software (v.3.6.2., www.r-project.org (accessed on 28 May 2021). A complete case analysis was undertaken (i.e., children with missing demographic, FMS, or SOCARP data were excluded from the analyses). Independent *t*-tests (age, BMI z-score, FMS outcomes), or chi-square tests (deprivation decile, ethnicity) were used to assess differences between those participants who were included and excluded from the final analysis.

A CoDA [46,47,52] was undertaken to examine associations between play behaviors and FMS. In advance, the cmultRepl function within the package zComposition (v.1.3.4) was used to replace zero counts in the play behavior data [53] before an orthogonal isometric logarithmic ratio (Ilr) transformation of the play variables. Descriptive statistics were subsequently calculated for the final sample: this included arithmetic means and standard deviations, geometric means for play behavior composite variables and pair-wise variation matrices to show the dispersion of the play behaviors (all calculated using the package ‘compositions’ v2.0.1). Each SOCARP play behavior category (*activity level*; *activity type*; *group size*) included time-use compositional data which was expressed as Ilr coordinates called pivot coordinates [47,54]. Specifically, *activity level* included five sets of four coordinates (time spent in lying, sitting, standing, walking, and very active); *activity type* included five sets of four coordinates (time spent in active games with equipment, active games without equipment, quiet play, sedentary, and locomotion); and *group size* included four sets of three coordinates (alone, small, medium, and large group).

To examine FMS associations with play behaviors, separate linear regression analyses were undertaken for each play behavior composition, i.e., *activity level*, *activity type*, and *group size*. Models included preschool as a random effect to account for the nesting of participants, and were adjusted for age, sex, and zBMI, but not deprivation, as this did not improve model fit. As a first step, the overall effect of the play behavior category composition was checked using the ANOVA table of model fit. If the play behavior composition was not significantly associated with the FMS outcome, no further analysis was undertaken. If the play behavior composition was significant, separate models were carried out using a different set of pivot coordinates, which encompass the full range of possible combinations of different behaviors relative to all the remaining behaviors in that category. Thus, equivalent statistical models were constructed for each play behavior category (e.g., *activity type*), with each variable within each set sequentially entered as the first Ilr coordinate (i.e., active games with equipment, active games without equipment, sedentary, quiet play, or locomotion), relative to all remaining play behavior variables in that category [46]. Total skill score, object-control, and locomotor skill scores represented the outcome (dependent) variables in the mixed linear regression models (run using the ‘stats’ package v3.6.2 and lm function), with the first isometric log-ratio coordinates (pivot coordinate) for each play behavior category entered as the explanatory (independent) variable [52]. The isometric log-ratio linear regression models were checked to ensure assumptions were not violated. Significance was set at *p* < 0.05.

## 3. Results

### 3.1. Descriptives

Table 1 shows descriptive statistics for the final sample with complete demographic, FMS, and SOCARP data, comprising 133 children aged 3–5 years (55% boys). Compositional variation matrices for play behavior data are in Appendix A. No significant differences were found for sex, zBMI, deprivation, or ethnicity between those included or excluded from the study, though those included were slightly older (*p* < 0.001). Many of the children (79.7%) lived in areas ranked in the highest decile for deprivation and were predominantly white British (82.7%), with the other children represented as mixed race (4.5%), other white descent (3.8%), Asian (3.8%), Black African (3.8%), or other (1.4%). Almost a quarter of the children (24.8%) were overweight or obese [51].

FMS competence scores were generally low-moderate across the sample, with locomotor scores higher than object-control skill scores. The most frequently observed play behavior activity level among the preschoolers was walking followed by standing and very active behaviors, while children spent less time in sitting and lying. Children therefore spent almost 70% of recess in moderate- to- vigorous-intensity physical activity. Children spent most of recess within small groups or on their own, rather than in medium or large groups. The most common activity types were active play with equipment and locomotion activities, followed by sedentary activities, with limited time spent in active play without equipment and quiet play.

### 3.2. Compositional Regression Analyses

Table 2 shows a summary of the isometric log-ratio regression models examining play behavior composites and FMS outcomes (total skills, object-control skills, and locomotor skills). Table 3 shows the subsequent further analyses examining the associations between total skill score and locomotor skill score FMS outcomes and play behavior activity type isometric log-ratio regression estimates (pivot coordinates).

#### 3.2.1. Total Skills and Play Behaviors

No significant associations were found for *activity level* and *group size* composite estimates with total skills score (Table 2). Therefore, no further analyses were undertaken for these play behavior compositions. *Activity type* was significantly associated with total skills score (Table 2), therefore further analyses were carried out. As shown in Table 3, relative to the other *activity type* behaviors, time spent in active games without equipment was positively associated with the total skills score (*β*_1_ = 2.03, *p* = 0.011), and time spent in locomotion activities was negatively associated with the total skills score (*β*_1_ = −2.96, *p* = 0.005).

#### 3.2.2. Object-Control Skills and Play Behaviors

No significant associations were found for *activity level*, *activity type*, and *group size* composite estimates with object-control skills score (Table 2). Therefore, no further analyses were undertaken.

#### 3.2.3. Locomotor Skills and Play Behaviors

No significant associations were found within *activity level* and *group size* composition estimates (Table 2). Therefore, no further analyses were undertaken for these play behavior compositions. *Activity type* was significantly associated with locomotor skills score (Table 2), therefore further analyses were undertaken. As shown in Table 3, relative to the other *activity type* behaviors, active games without equipment were positively associated with locomotor skills (*β*_1_ = 1.08, *p* = 0.013), and time spent in locomotion was negatively associated with locomotor skills (*β*_1_ = −1.50, *p* = 0.009).

## 4. Discussion

This study aimed to examine the relationships between FMS and play behaviors in typically developing preschool children during preschool recess using CoDA. Significant associations were observed within play behavior activity type and FMS. Time spent in active games without equipment, relative to other activity types, was positively associated with higher total skill and locomotor skills scores. Time spent in locomotion (moving while not engaged in an active play game, e.g., transitions from one play activity to the next), relative to the other activity types, was negatively associated with total skill score and locomotor skill score. No associations were observed for activity intensity level or group size play behavior time-use composites with FMS. The findings indicate that participation in specific types of play behaviors during recess are potentially important for FMS development in young children. This is the first study to use CoDA to examine FMS and play behaviors in preschool children, a statistical approach which recognizes that time dependent behaviors (e.g., during recess) are mutually exclusive, as time spent in one behavior can only be changed by concurrently changing one or more other behaviors by the same duration [46,47]. The findings from previous literature are predominantly based on univariate analysis where behaviors are analyzed in isolation from the remaining behaviors [47]. While not directly comparable to the present study, these studies are incorporated into the discussion to facilitate the interpretation and explanation of the results.

A key finding in the present study was that time spent in active games without equipment, relative to the other activity types, was positively associated with total and locomotor FMS scores. Though, on average, children in our sample spent limited time in this type of activity, this finding suggests that spending more time on active games without equipment such as dancing, hide and seek, chasing games, imaginative play, rough and tumble, as well as verbal games that involve actions and clapping (e.g., ring-a-roses), may be important for FMS development. Alternatively, children with high FMS competence spend a higher proportion of their recess playtime participating in active games without equipment. It can be expected that these activities are related to locomotor skills, as running, hopping, jumping, leaping, galloping, and sliding are frequently utilized in these types of play. Previous research has demonstrated that participation in dance activities during the preschool day is positively associated with locomotor skill development in young children [55]. However, counterintuitively, no relationship was observed in the study between frequency of walking or running activities and locomotor skills [55]. The present study included locomotion as an activity type, which represented children engaged in a locomotor activity (e.g., walking, jogging, running, skipping without a rope) that *was not* part of a sport or active game, such as while transitioning from one activity to the next. These locomotion activities were negatively associated with total and locomotor skill scores. This may represent children that are on the periphery of participating in active game play and struggling to find an engaging and meaningful play activity that might support skill development—a phenomena described by Herrington and Brussoni as ‘channel surfing’ [56]. These children may require encouragement and need to be offered a range of possibilities to substitute this locomotion activity with more active forms of play. Indeed, it is possible that simply moving or transitioning from one place to another without an active play purpose may not be sufficient to foster FMS; meaningful, playful locomotor activities may be necessary needed to acquire skills.

On average, children spent a relatively large proportion of recess time (41%) engaged in active games with equipment, yet greater time spent in this type of play behavior was not associated with FMS. This may suggest that the fixed and portable (loose parts) equipment available in these preschool settings did not provide children with the affordances for locomotor and object-control skill development. Like Tsuda et al., who examined FMS and physical activity during free play in two preschools [57], our observations revealed that limited bats and balls were available during recess for children to practice object-control skills. Children were frequently observed idly sitting on wheeled toys or aboard fixed climbing structures. It is possible that these pieces of equipment supported other FMS capacities, such as lower body strength, climbing, or stability skills, not assessed in the present study. Nevertheless, our finding is similar to previous research that reported that different types of playground design and equipment involved a limited number of FMS [58]. A systematic review examining the value of playgrounds for children’s physical activity levels found that the presence of a fixed structure athletics track was positively associated with physical activity, while the availability of slides, sandboxes, and swinging equipment on the playground—all of which can involve turn-taking—were negatively associated with activity levels [59]. Other studies have found less fixed or static playground equipment and more portable play equipment (e.g., balls, portable slides) to be beneficial for young children’s physical activity levels [60,61,62]. While this suggests that upgrading the type and volume of portable equipment in preschools could assist with engaging children in playful activities that facilitate the development of object-control skills (e.g., providing a wide variety and number of balls in different sizes and colors), it is recommended that future research examines the association between FMS and fixed versus portable equipment separately. It is important to note that the above-mentioned studies investigated physical activity levels rather than FMS and focused on the presence of equipment in the playground, rather than children’s engagement in active games using the equipment like in the current study. Nevertheless, these studies and others highlight numerous preschool and playground environmental characteristics that contribute to children’s activity levels at preschool such as larger playground size, presence of an open field with no markings, and fewer children on the playground [59]. Thus, given that physical activity drives FMS development in the early years [2], further research examining the influence of preschool physical environmental characteristics and the volume and type of fixed and portable equipment on active play and FMS is warranted.

Geometric means indicated that participants in this study spent relatively greater recess time in moderate- to- vigorous physical activity, comprising walking (40%) and very active behaviors (28%). Recess periods in preschool are shorter and more frequent, and there is evidence that this leads to increased moderate- to- vigorous physical activity [39,63], which may explain the very active physical activity levels observed. However, findings showed that the activity intensity play behavior composite mean was not associated with preschoolers’ FMS. This finding is somewhat inconsistent with recent evidence from two systematic reviews that found positive associations between physical activity levels and FMS among young children, including at moderate- and- vigorous intensities [23,24]—inclusive of the results from our own research [22], which involved the same sample of children involved in the current study. Our study and others included in the systematic reviews examined habitual physical activity and FMS. Results from studies examining FMS and young children’s physical activity during preschool hours and specifically preschool recess are mixed. For example, Iivonen et al. [64] directly observed physical activity during three consecutive preschool days in a small sample of Finnish children (n = 53) and found that light and moderate-to-vigorous physical activity were not associated with FMS. In contrast, Tsuda et al. [57] examined physical activity using accelerometers during free-play time at preschool (i.e., recess) in a cross-sectional study and reported that locomotor and object-control skills significantly predicted moderate- to- vigorous physical activity (n.b., the authors did not examine FMS as the dependent variable). To cloud the issue further, there is evidence that suggests that low intensity activities at preschool are associated with FMS. Martins et al. recently demonstrated through accelerometry and CoDA that increasing sedentary time at the expense of light physical activity elicited improved manipulative skills in preschool children [65]. In relation, Butcher and Eaton [66] found that 5-year-old children who participated in low intensity, fine motor activities during indoor free play were more likely to have good visual motor control and balance. Taken together, these diverse results indicate that more research is needed to examine the relationship between activity intensity and FMS during preschool, and specifically during recess. Studies that capture information about the types and context of physical activity during recess alongside the intensity of movement are needed to better understand the nature of playful skill development.

No associations were observed for group size play behavior composites with FMS. On average, children spent a relatively large proportion of recess time playing alone or in small groups. Neither individual or group activity was found to be important for FMS, but this may change over time as children’s social and emotional skills develop and their play preferences mature from solitary play to complex social play [19,67]. For example, ball skills may be augmented through small-sided games as children progress to more stereotypical playground activities in primary school that involve larger group sizes such as football. We did not examine whether the composition of groups (e.g., same sex versus mixed), teacher involvement, or teacher proximity to child play activities were associated with preschoolers’ FMS development. Herrmann et al. [19] have described how young children tend to make friends with the same gender. Furthermore, boys engage in more individual play and girls more frequently engage in cooperative play. Thus, gender differences in play group compositions may influence FMS. Previous studies have also shown how childcare educators’ education, attitudes, beliefs, behaviors, and practices may influence preschool children’s physical activity, FMS, and physical literacy [68,69,70]. Therefore, examining if and how teacher attitudes and practices in relation to play behaviors and FMS and how teacher interactions with children during recess affect FMS competence could be an interesting area for future study.

The purpose of the current study was to better understand how young children’s preschool recess play behaviors are related to FMS competence. Under the United Nations Convention on the Rights of the Child, children have the right to play [71]. It is important to emphasize that the authors’ position is that recess play should remain play, i.e., freely chosen, purposeless, self-directed, and intrinsically motivated [29,30,31]. Though we were interested in play and FMS development, we consider that play is an end in itself [56], and recess play to be a context for activities that are unstructured and fun. We also recognize that play is a diverse and complex behavior that is essential for child development [72]. Thus, we are not advocating for adult-directed, structured FMS programs during preschool recess, for example, to deliver active games without equipment. Rather, preschool settings and educators should seek to maximize the opportunities for young children to engage in diverse active play experiences [73]. The environmental resources available to each child to foster FMS could be enhanced through changing the design of play spaces within the recess playground to include natural landscapes and features (e.g., forested areas with trees and shrubs, rocks, water, sand, uneven ground, slopes) [56,74,75], as well as increasing the volume and range of loose parts and fixed equipment [60,61,62]. The role of preschool educators should be to encourage and offer possibilities for active play while fully respecting child agency [68,69,70]. Nevertheless, due to the narrow evidence base, more research exploring how to maximize affordances for young children to develop FMS during preschool recess periods is warranted.

The strengths of this study include the use of direct observation and video assessments of FMS and play behaviors, which ensured that information about the quality of FMS movements and types of play behaviors (rather than just activity levels) were captured. Furthermore, recruitment included a representative sample from northwest England, which is more deprived than other parts of the country. A major strength is the use of compositional data analyses to consider the time dependent nature of the play behavior data to examine the associations between play behaviors, relative to one another, with FMS. The limitations of the study include a lack of generalizability of the study findings due to the primarily disadvantaged and regional sample. Further, play behavior data was captured through 5-min observations. Longer observation periods may have captured more diverse play behaviors. In addition, the time of day of the recess periods might have also influenced play behaviors but was not computed for use in the analysis. Furthermore, there was a high proportion of missing data as feasibility constraints meant that SOCARP measurements could only be captured in a sub-sample of children, while the total number FMS assessments were limited by absent children or missing skills due to technical issues. Furthermore, capturing information about the environmental characteristics and policies in preschool settings that may influence FMS affordances and physical activity (such as through the Environmental and Policy Assessment and Observation Tool: EPAO [76]) would have facilitated a deeper understanding and stronger interpretation of the study findings had they been measured and controlled for. Similarly, SOCARP captures broad activity types and coding play behaviors in a more detailed way, such as using the recently developed Tool for Observing Play Outdoors (TOPO) [77], may help to facilitate a more detailed understanding of the association between play activity types and FMS, such as the specific and different types of active games with and without equipment. Finally, the study only included assessments of locomotor and object-control FMS. Had a broader range of FMS assessments been used, such as stability and fine motor skills as well as broader FMS such as strength or cycling, more associations with play behaviors could have been uncovered.

## 5. Conclusions

In conclusion, significant associations were found within the play behavior activity type compositions: relatively more time spent in active games without equipment were associated with higher total and locomotor FMS scores, while relatively more time in locomotion activities was associated with lower total and locomotor FMS scores. No associations were found between activity level and group size play behavior compositions and FMS. The findings indicate that participation in specific play behaviors during recess may be important for FMS development in young children, though we are unable to draw causal conclusions. Future research, including longitudinal data, is required to confirm and expand these findings.

## Figures and Tables

**Table 1 children-08-00543-t001:** Descriptive statistics for the final sample (*n* = 133).

Variables	Mean (SD)	Geometric Mean
*Demographics*		
Age (Years)	4.70 (0.53)	-
Body mass index z-scores	0.74 (1.00)	-
*Play Behaviors*		
*Child Activity Level*		
Lying (%)	0.73 (3.51)	0.11
Sitting (%)	8.04 (14.50)	3.43
Standing (%)	27.81 (19.01)	28.30
Walking (%)	33.85 (17.02)	40.14
Very Active (%)	29.58 (21.29)	28.02
*Activity Type*		
Active Games with Equipment (%)	38.24 (38.89)	41.11
Active Games without Equipment (%)	11.44 (20.47)	5.85
Sedentary (%)	14.93 (18.80)	13.60
Quiet Play (%)	8.37 (19.38)	3.23
Locomotion (%)	27.02 (25.08)	36.21
*Group Size*		
Alone (%)	37.64 (32.86)	33.99
Small (%)	52.49 (31.30)	62.48
Medium (%)	8.97 (19.64)	3.38
Large (%)	0.90 (5.30)	0.15
*Foundational Movement Skills*		
Total score (range: 0–142 skill components)	63.04 (12.68)	-
Object control skills (range: 0–78 skill components)	29.47 (8.04)	-
Locomotor skills (range: 0–64 skill components)	33.57 (6.74)	-

**Table 2 children-08-00543-t002:** Isometric log-ratio regression models examining play behavior composites and FMS.

	Total Skills Score			Object-Control Skills			Locomotor Skills		
Model	*X* ^2^	df	Pr (>*X*^2^)	*X* ^2^	df	Pr (>*X*^2^)	*X* ^2^	df	Pr (>*X*^2^)
*Activity Level*									
Activity Level Ilr	7.66	4	0.105	9.04	4	0.060	3.04	4	0.551
Body mass index	1.51	1	0.219	1.73	1	0.189	0.63	1	0.428
Sex	0.72	1	0.397	10.04	1	0.002 *	4.48	1	0.034 *
Age	19.30	1	0.000 *	16.36	1	0.000 *	12.48	1	0.000 *
*Activity Type*									
Activity Type Ilr	11.66	4	0.020 *	7.80	4	0.092	9.93	4	0.042 *
Body mass index	3.69	1	0.054	4.00	1	0.046 *	1.47	1	0.225
Sex	0.08	1	0.782	6.85	1	0.009 *	6.27	1	0.012 *
Age	15.32	1	0.000 *	11.55	1	0.001 *	10.06	1	0.002 *
*Group Size*									
Group Size Ilr	0.84	3	0.840	0.85	3	0.836	1.51	3	0.679
Body mass index	2.81	1	0.093	2.84	1	0.092	1.35	1	0.245
Sex	0.29	1	0.593	7.63	1	0.006 *	5.11	1	0.024 *
Age	15.61	1	0.000 *	13.26	1	0.000 *	10.38	1	0.001 *

*Notes*. Ilr = play behavior composite variable; *X*^2^ = chi-square value; df = degrees of freedom; Pr (>*X*^2^) = probability of observed chi-square statistic that indicates whether the regression coefficient is not equal to zero in the model; * significant association with the FMS outcomes (*p* < 0.05). All models included preschool as a random factor to account for nesting of participants.

**Table 3 children-08-00543-t003:** Summary of associations between FMS outcomes and play behavior activity type isometric log-ratio regression estimates (pivot coordinates).

	Total Skills Score				Locomotor Skills			
*Model* First Pivot Coordinate	*β*_1_ Ilr	LCI	UCI	*p*-Value	*β*_1_ Ilr	LCI	UCI	*p*-Value
*Activity Type*								
(i) Active gameswith equipment	−0.87	−2.94	1.20	0.412	−0.08	−1.20	1.04	0.884
(ii) Active gameswithout equipment	**2.03**	**0.46**	**3.60**	**0.011 ***	**1.08**	**0.23**	**1.93**	**0.013**
(iii) Sedentary	0.51	−1.17	2.20	0.550	0.44	−0.47	1.35	0.342
(iv) Quiet Play	0.13	−1.28	1.55	0.853	0.30	−0.46	1.07	0.438
(v) Locomotion	**−2.96**	**−5.02**	**−0.89**	**0.005 ***	**−1.50**	**−2.62**	**−0.37**	**0.009**

*Notes*. *β*_1_ Ilr = first isometric log-ratio regression coefficients (pivot coordinate), which should be considered in terms of reallocating time to the behavior relative to the remaining activity type behaviors; LCI: lower 95% confidence interval limit; UCI: upper 95% confidence interval limit. Separate models were run for each set of pivot coordinates. All models included school as a random factor and were adjusted for age, sex, and zBMI. * Bolded coefficients = significant association with the FMS outcomes (*p* < 0.05).

## Data Availability

The data presented in this study are available upon request from the corresponding author. The data are not publicly available due to ethical approval restrictions.

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
