# Peer review of "Foundational Movement Skills and Play Behaviors during Recess among Preschool Children: A Compositional Analysis"

_children, 2021, doi:10.3390/children8070543_

Round 1
Reviewer 1 Report
This is an interesting paper, using a novel approach to investigate the association between play behaviour and the quality of motor competence in early childhood. The paper is well-written and has a logic progression. Although I am not an expert in this type of analysis, the method is easy to follow and seems sound. I have only a few minor comments:
Title: You use the term foundational motor skills (FMS), while others have used fundamental movement skills. I wonder whether you see a difference between the two, or why you have opted for "foundational". It might be worth clarifying your choice to the readers.
L105: The fact that the study focused on preschools within disadvantaged communities is presented as a strength at the end of the discussion. While I agree that this may be representative for parts/regions within the Northwest, this is clearly also a limitation. I would appreciate some appraisal of this issue.
L161: Please clarify who coded the videos. Also, is my understanding correct that you have 5 minutes of footage per child. This may be common in this type of research, however, it is also a limitation. Have you checked if there was an effect of the timing of recess. I can imagine that some children are somewhat withdrawn during the morning session, after they have been dropped off.
L166 You refer to training, but who was this training for. It sounds like the person who did the coding for the inter-rater analysis was trained, but I would imagine that whoever did the coding needed dedicated training. Please clarify
L189: Please write the part between brackets in full. "This included calculation of the arithmetic means and standard deviations, the geometric means and...
L194: It is not clear to me what is meant with five sets of four coordinates, but that is probably due to my limited understanding of the CoDA.
L268: Delete "To" at the start of the sentence
L279-281: This conclusion is not in line with your findings, please rephrase. We can observe a number of interesting associations, but concluding that there is an important causal relationship between play and FMS development is an overstatement.
L323: Here, I wondered whether you have / could provide information on the availability of age/developmentally appropriate equipment in the preschools, especially given the fact that the schools are in rather deprived areas. This is addressed to some extent in the following paragraph but it would be relevant to integrate this here already.
L436: with should be without here.
Supplementary material: Again due to my lack of knowledge of CoDA I have difficulty with understanding the matrices. Would it be possible to add what the coefficients mean in the description of the table?
On a more general and philosophical note, I really believe that play should remain play, and therefore consist of freely chosen activities. The task of schools / teachers is in providing the right context (space, equipment, etc.), and maybe to facilitate activity and skill development through nudging / or encouragement, not via instruction, otherwise the recess is turned into a (PE)class. I don't have the impression that you are of a different opinion, but it may be worth highlighting this in the discussion.
Author Response
|
# |
Comment |
Authors’ response |
|
1 |
This is an interesting paper, using a novel approach to investigate the association between play behaviour and the quality of motor competence in early childhood. The paper is well-written and has a logic progression. Although I am not an expert in this type of analysis, the method is easy to follow and seems sound. I have only a few minor comments:
|
Thank you for your kind feedback on the manuscript and useful comments. |
|
2 |
Title: You use the term foundational motor skills (FMS), while others have used fundamental movement skills. I wonder whether you see a difference between the two, or why you have opted for "foundational". It might be worth clarifying your choice to the readers.
|
This was a deliberate decision, and we are happy to clarify this further. Previously, we would have used fundamental movement skills, but we wholeheartedly agree with the conceptual adaptation proposed by Hulteen et al. and want to support a transition to this term within the literature. We particularly support this given the contentious debate around the term ‘fundamental’ in fundamental movement skills, which has led some in the field to question the importance of these skills. As stated by Hulteen et al.
“When something is ‘fundamental’, that means it is of, or relating to, necessary structure or function. Thus, the capability to competently kick, catch, run, or jump can be implied to mean necessary for physical activity participation. While developing competency in these skills certainly would facilitate physical activity participation, a lack of competency in one skill (e.g., kicking) doesn’t necessarily result in inactivity. Rather, that individual may have fewer physical activity options compared with someone who demonstrates competency in that particular skill. Meanwhile, ‘foundational’ refers to ‘an underlying base or support’. Thus, the development of foundational movement skill competency will, just as motor development models show, support and maximize opportunities for participation in physical activity. …The term ‘foundational movement skills’ better reflects the broad range of movement forms that increase in complexity and specificity and can be applied in a variety of settings. Thus, ‘foundational movement skills’ includes both traditionally conceptualized ‘fundamental’ movement skills and other skills (e.g., bodyweight squat, cycling, swimming strokes) that support physical activity engagement across the lifespan.”
For brevity reasons, we have added a short section to the opening paragraph to expand on foundational movement skills (see Lines 34 to 40).
“FMS is a relatively new term that includes both traditionally conceptualized fundamental movement skills such as stability (e.g., sitting, standing, balancing on a foot), locomotor (e.g., running, jumping, crawling) and object control (e.g., striking, catching, throwing) skills, and other skills that support lifelong engagement in physical activity (e.g., squat, cycling, swimming). ‘Foundational’ refers to these skills providing an ‘underlying base or support’, with the development of greater competency in many skills providing more options for physical activity across the life course [1]. |
|
3 |
L105: The fact that the study focused on preschools within disadvantaged communities is presented as a strength at the end of the discussion. While I agree that this may be representative for parts/regions within the Northwest, this is clearly also a limitation. I would appreciate some appraisal of this issue.
|
Thank you for highlighting this. We have amended the limitations section to note the lack of generalizability of the findings due to the disadvantaged study sample. (see Line 440). |
|
4 |
L161: Please clarify who coded the videos. Also, is my understanding correct that you have 5 minutes of footage per child. This may be common in this type of research, however, it is also a limitation. Have you checked if there was an effect of the timing of recess. I can imagine that some children are somewhat withdrawn during the morning session, after they have been dropped off.
|
The play behavior videos were coded by MOD (see Line 168). This has been clarified in the text. We have added the length of the play observation time window as a study limitation. Unfortunately, we did not record the data on the timing of recess and so have also added this to the study limitations. (see Line 441) |
|
5 |
L166 You refer to training, but who was this training for. It sounds like the person who did the coding for the inter-rater analysis was trained, but I would imagine that whoever did the coding needed dedicated training. Please clarify
|
The previous line stated that coding was completed by a trained observer (MOD). We have clarified in the subsequent sentence that “ ‘Observer’ training was conducted…“ (Line 168) |
|
6 |
L189: Please write the part between brackets in full. "This included calculation of the arithmetic means and standard deviations, the geometric means and...
|
Thank you – we have revised this sentence and removed parts of this statement from the brackets to improve readability. (Lines 191-195) |
|
7 |
L194: It is not clear to me what is meant with five sets of four coordinates, but that is probably due to my limited understanding of the CoDA.
|
It is quite a difficult and challenging analysis to explain and understand. The later section of the stats analysis section tried to explain the nature of pivot coordinates for the reader (see Lines 209 to 215). We have tried to be concise within the manuscript - readers can refer to the supporting references for further explanations of CoDA.
|
|
8 |
L268: Delete "To" at the start of the sentence
|
Thank you for spotting this error at the start of the discussion, which we have corrected.
|
|
9 |
L279-281: This conclusion is not in line with your findings, please rephrase. We can observe a number of interesting associations, but concluding that there is an important causal relationship between play and FMS development is an overstatement.
|
We’re not convinced that we implied that there was a causal relationship in our conclusion text. Nevertheless, to avoid any doubt, we have extended the final sentence to clarify that we cannot draw causal conclusions from this data and more research, including longitudinal research is needed. (see Lines 468-470)
|
|
10 |
L323: Here, I wondered whether you have / could provide information on the availability of age/developmentally appropriate equipment in the preschools, especially given the fact that the schools are in rather deprived areas. This is addressed to some extent in the following paragraph but it would be relevant to integrate this here already.
|
After re-running the analysis following the comment from Reviewer 2 to include school as a nesting variable, we have deleted this section of text as the finding of active games without equipment being associated with object-control skills was no longer significant.
|
|
11 |
L436: with should be without here.
|
Thank for highlighting that mistake in the conclusion, which we have now corrected.
|
|
12 |
Supplementary material: Again due to my lack of knowledge of CoDA I have difficulty with understanding the matrices. Would it be possible to add what the coefficients mean in the description of the table?
|
Thank you for the suggestion. We have added a footnote to the supplementary tables to describe what the variation matrices mean, i.e. that a value close to zero implies that the two parts involved in the ratio (arranged by the rows and columns in the matrix) are highly proportional.
|
|
13 |
On a more general and philosophical note, I really believe that play should remain play, and therefore consist of freely chosen activities. The task of schools / teachers is in providing the right context (space, equipment, etc.), and maybe to facilitate activity and skill development through nudging / or encouragement, not via instruction, otherwise the recess is turned into a (PE)class. I don't have the impression that you are of a different opinion, but it may be worth highlighting this in the discussion. |
We completely agree with the reviewer and have added a paragraph to the discussion to that effect. See lines 413-432.
“The purpose of the current study was to better understand how young children’s preschool recess play behaviors are related to FMS competence. Under the United Nations Convention on the Rights of the Child, children have the right to play [71]. It is important to emphasize that the authors’ position is that recess play should remain play, i.e., freely chosen, purposeless, self-directed, and intrinsically motivated [29-31]. Though we were interested in play and FMS development, we consider that play is an end in itself [56], and recess play to be a context for activities that are unstructured and fun. We also recognize that play is a diverse and complex behavior that is essential for child development [72]. Thus, we are not advocating for adult-directed, structured FMS programs during preschool recess; for example, to deliver active games without equipment. Rather, preschool settings and educators should seek to maximize the opportunities for young children to engage in diverse active play experiences [73]. The environmental resources available to each child to foster FMS could be enhanced through changing the design of play spaces within the recess playground to include natural landscapes and features (e.g., forested area with trees and shrubs, rocks, water, sand, uneven ground, slopes) [56, 74, 75], as well as increasing the volume and range of loose parts and fixed equipment [60-62]. The role of preschool educators should be to encourage and offer possibilities for active play, while fully respecting child agency [68-70]. Nevertheless, due to the narrow evidence base, more research exploring how to maximize affordances for young children to develop FMS during preschool recess periods is warranted.”
|

Reviewer 2 Report
Interesting paper. It is important to get more knowledge on play behavior and motor skills/development. The paper is well written and has good consistence and structure.
The introduction provide sufficient background, include relevant references and give a good rationale for the study.
Methods: Measures are apropriate. The data is quite old, but I presume it doesn't influence the aim and research question. Also, the response rate is quite low, but the authors do point out this in the paper as a limitation. I miss a better description of the context in which the play behavior was measured (point 2.3.2). Actually this is a critical point for the study. With 12 different ECECs one can assume the play environment and facilities differs quite much (small spaces could restrict mobility etc.). This could affect play behavior as much as FMS. I would be happy if the authors commented to this. Also, was this data collection outdoors only?, could be stated (or maybe I have overlooked the information).
I'm not familiar with the CODA, however, the result section is fine.
Discussion section is well written and easy to follow. Conclusions are supported by the results.
Check typo: line 268
Reference 26 and 28 seems to be the same.
Author Response
|
# |
Comment |
Authors’ response |
|
1 |
Interesting paper. It is important to get more knowledge on play behavior and motor skills/development. The paper is well written and has good consistence and structure. |
Thank you for your kind feedback and helpful comments below. |
|
2 |
The introduction provide sufficient background, include relevant references and give a good rationale for the study.
|
Thank you. |
|
3 |
Methods: Measures are appropriate. The data is quite old, but I presume it doesn't influence the aim and research question. Also, the response rate is quite low, but the authors do point out this in the paper as a limitation. I miss a better description of the context in which the play behavior was measured (point 2.3.2). Actually this is a critical point for the study. With 12 different ECECs one can assume the play environment and facilities differs quite much (small spaces could restrict mobility etc.). This could affect play behavior as much as FMS. I would be happy if the authors commented to this. Also, was this data collection outdoors only?, could be stated (or maybe I have overlooked the information).
|
Thank you for these important considerations. We have clarified that the play observations took place outdoors on the preschool area (see Line 165). We did not capture and record information about the preschool ECECs environmental characteristics and policies and mentioned this as a limitation in the original manuscript (see Line 448).
Importantly, the reviewer’s comment did make us reflect on our statistical analyses. We did not account for the nesting of participants within schools within our original statistical analysis. Doing so would help to control for preschool related effects on the associations due to factors highlighted by the reviewer. Therefore, we re-ran our statistical models and included school as a random factor to account for nesting. The re-analysis led to some significant changes in our results, but the data makes more sense now. The only significant associations found were in relation to activity type play behavior compositions and FMS. No other associations were found. We have amended the results text and discussion text accordingly. We thank the reviewer for highlighting this important point and we feel that we now have more robust statistical analyses and findings.
|
|
4 |
I'm not familiar with the CODA, however, the result section is fine.
|
Thank you. |
|
5 |
Discussion section is well written and easy to follow. Conclusions are supported by the results.
|
Thank you – as noted above, we have amended the discussion following the changes in findings because of re-running the statistical analysis to account for nesting. Please see track changes in the revised manuscript.
|
|
6 |
Check typo: line 268
|
Corrected, thank you. |
|
7 |
Reference 26 and 28 seems to be the same.
|
Corrected, thank you. |
